# Assessing the Spatial Variability of Daytime/Nighttime Extreme Heat Waves in Beijing under Different Land-Use during 2011–2020

Xiaokang Su [1,2], Fang Wang [2,*], Demin Zhou [1] and Hongwen Zhang [3]

[1] College of Resource Environment and Tourism, Capital Normal University, Beijing 100048, China
[2] Key Laboratory of Land Surface Pattern and Simulation, Institute of Geographic Sciences and Natural Resources Research, Chinese Academy of Sciences, Beijing 100101, China
[3] Beijing Meteorological Disaster Prevention Center, Beijing 100089, China
[*] Correspondence: wangf@igsnrr.ac.cn; Tel.: +86-010-64889829

**Abstract:** Urban land-use affects surface air temperature; however, the impact of urban land-use on surface air temperature, particularly the extent to which it affects the duration of extreme heat waves, remains uncertain and the mechanisms of diurnal differences need to be further explored. This paper presents study of daytime/nighttime extreme heat waves duration in Beijing under different land-use changes by adopting an index of cumulative hours of extreme heat waves exceeding the certain thresholds. The urban day/night extreme heat waves cumulative hourly interpolation models were established based on high-resolution urban land-use and socioeconomic data and were assessed to have good performance. The annual average cumulative hours of extreme heat waves increased by 95% (daytime) and 116% (nighttime) in 2016–2020 compared to 2011–2015. The cumulative hours for each land-use type ranked as follows: urban land > cropland > water > grassland > woodland. We found that the cumulative hours of extreme heat waves increased significantly with the proportion of urban land and decreased significantly with the proportion of forested land and water. This research provided important information for alleviating extreme heat waves in cities and for rational land planning.

**Keywords:** extreme heat waves; land use/cover; interpolation models; daytime; nighttime

## 1. Introduction

Under the background of global climate change, the frequent occurrence of extremely high temperature weather has a great impact on human health and social economy [1,2]. Studies have shown that land use/cover changes alter the surface roughness, vegetation coverage, and surface reflectivity, which directly affect the latent heat, sensible heat, and water exchange between the surface and the atmosphere, thereby affecting the local regional climate and promoting thermal environment changes [3,4]. Urban land expansion is closely related to urban high temperatures [5–7], and urban land use expansion changes the thermodynamic characteristics of the underlying surfaces of urban areas to a large extent, and also has a certain impact on the global temperature increase [8,9]. Therefore, land-use change can be regarded as the main factor affecting urban high temperatures [10].

Some studies explored the relationship between urban land-use change and urban land surface temperature from different perspectives. Studies were made of the impact of urban expansion on urban land surface high temperatures by analyzing the relationships between various types of building indices or construction land scale and surface temperature [11–13]. Some studies analyzed the relationships between changes of different land-use types and surface temperature [14,15] or usage of remote sensing indices (Normalized Difference Vegetation Index (NDVI), Normalized Difference Water Index (NDWI), etc.) to conduct correlation with surface temperature [16,17]. Some scholars estimated the contribution

rates of different land-use types to high temperatures [18–21]. In addition, related research introduced the landscape pattern index to study their impact on urban land surface high temperatures [22,23]. In general, most of the current studies have focused on surface temperature rather than surface air temperature. However, the impact of urban land-use on surface air temperature, especially the extent of the effect on the duration of extreme heat waves, remains uncertain, and the mechanisms of their diurnal differences need to be further explored.

In this study, we explored daytime/nighttime extreme heat waves changes in Beijing under different land-use changes from 2011 to 2020 spanning the 12th Five-Year Plan and 13th Five-Year Plan. Data measured by 225 meteorological stations in the Beijing area is used in this study and are provided by the Beijing Meteorological Disaster Prevention Center. The 90% quantile was taken as the high temperature threshold, and the cumulative hours of extreme heat waves exceeding the thresholds (daytime: 33.1 °C; nighttime: 27.9 °C) were adopted to characterize the intensity and duration of extreme heat waves. We established the urban day/night extreme heat waves interpolation models based on high-resolution urban land use and socioeconomic data during 2011–2020. The relationship between different land-use changes and accumulated hours of day/night extreme heat waves was explored.

## 2. Materials and Methods

### 2.1. Study Area

Beijing is located within the range of 115.7° E–117.4° E, 39.4° N–41.6° N, with an altitude range of 6–2300 m. It has a jurisdiction with over 16 districts and 331 townships, covering an area of 16,410.54 km$^2$ (Figure 1a). At the end of 2020, Beijing's permanent population was 21.893 million of which the urban population was 19.166 million, accounting for 87.5%. The gross domestic product (GDP) of Beijing is 3.6 trillion yuan, and the city is worthy of being called an international metropolis and representative of China's rapid urbanization (Figure 1b). The region has a typical temperate sub-humid continental monsoon climate with high temperatures and a rainy summer. Through calculation, we found that, from 2011 to 2020, the average temperature in summer (June–August) in Beijing increased at a rate of 0.64 °C/10a, and the number of hot days also increased, making Beijing one of the hottest cities in China.

### 2.2. Data Collection and Preprocessing

#### 2.2.1. Meteorological Data

According to the official data published by the China Meteorological Administration, the average daily maximum temperature in Beijing was significantly higher in June, July, and August than in other months. Therefore, in this study, summer was chosen as the time period when extreme heat waves occur frequently in Beijing. Hourly temperature data from Beijing meteorological stations (MSs) from June to August during 2011–2020, provided by the National Meteorological Information Center, were used in this study. We selected a total of 225 MSs (20 national weather stations-NWSs, 205 automatic weather stations-AWSs) that were established before June 2011 and still in use until September 2020 (Figure 1a). Most NWSs are responsible for regional or national weather information exchange and are the main body of the national weather and climate website. The observation data obtained by AWS are mainly used for weather services in their own provinces (districts and cities) and localities and also complement the observation data on the national weather and climate website. All observation data were processed by the National Meteorological Information Center for strict quality control. The outliers in the original data were eliminated, and the missing data are replaced by observations at adjacent times or average values, reducing the error caused by instrument failure or measurement error, and the data are highly accurate.

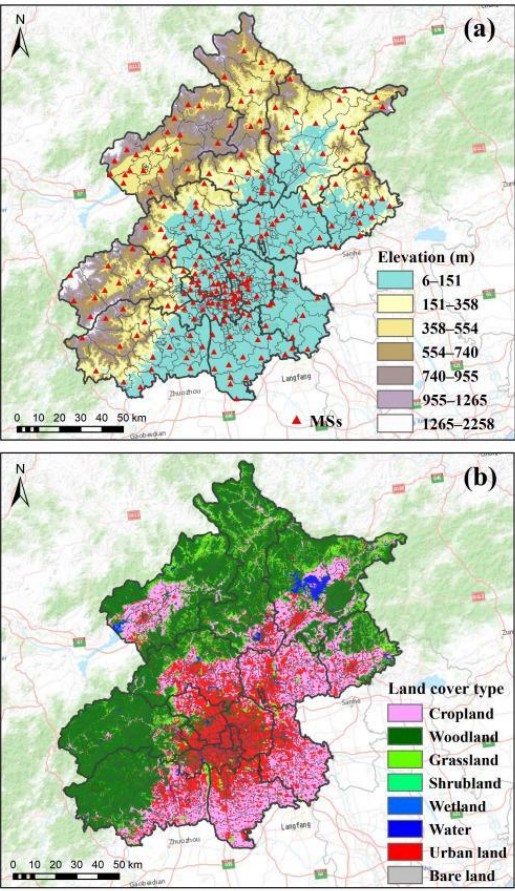

**Figure 1.** (**a**) Elevations and locations of meteorological stations (MSs). (**b**) Land cover types of cropland, woodland, grassland, shrubland, wetland, water, urban land, and bare land in 2020 (The black border represents the administrative border of the district; the gray border represents the administrative border of the township. Data on administrative boundaries were obtained from the National Geomatics Center of China).

### 2.2.2. Land Cover

Land cover data were from the GlobeLand30 V2010 and GlobeLand30 V2020 database (downloaded at http://www.globallandcover.com (accessed on 8 July 2022)), with a spatial resolution of 30 m, include 10 types, namely: cropland, woodland, grassland, shrubland, wetland, water, tundra, urban land, bare land, and glacier. The data accuracy of GlobeLand30 V2010 was evaluated by Tongji University. Over 150,000 test samples were laid down, and the overall accuracy of GlobeLand30 V2010 data was 83.50% with a Kappa coefficient of 0.78. The data accuracy of GlobeLand30 V2020 was evaluated by the Aerospace Information Research Institute, Chinese Academy of Sciences. With over 230,000 samples laid down, the overall accuracy of GlobeLand30 V2020 data was 85.72% with a Kappa coefficient of 0.82. The Kappa coefficient is used for consistency testing and can also be used to measure classification accuracy. In general, a higher Kappa coefficient indicates a higher classification accuracy.

### 2.2.3. Population Density

The spatial distribution kilometer grid data of China's population density in 2015 and 2020 were used, which were from the Institute of Geographic Sciences and Natural Resources Research (IGSNRR, https://www.resdc.cn/ (accessed on 8 July 2022)), with a resolution of 1 km. The population density unit is person/km². The cumulative population of each district in Beijing was calculated by the ArcGIS statistical tool and verified by the

resident population of each district in Statistical Yearbook data. The two showed a strong correlation, with $R^2$ values of 0.99 (2015) and 0.99 (2020).

### 2.2.4. GDP

Spatial distribution kilometer grid data of China's GDP in 2020 from the IGSNRR were used, with a resolution of 1 km. Each grid cell represents the total GDP of the area in units of 10,000 yuan/km$^2$. The cumulative GDP of each district in Beijing was calculated using the ArcGIS zoning statistical tool and verified using the total GDP of each district in the Beijing Regional Statistical Yearbook 2021 (http://nj.tjj.beijing.gov.cn/nj/qxnj/2021/zk/indexch.htm (accessed on 8 July 2022)). The two showed a strong correlation with $R^2$ values of 0.96 (2015) and 0.96 (2020).

### 2.2.5. DEM

Digital Elevation Model (DEM) data were derived from the Shuttle Radar Topography Mission (SRTM) data of the US Space Shuttle Endeavor. In this study, the latest SRTM V4.1 data were resampled to generate a new dataset with a resolution of 250 m. Data were projected using a WGS84 ellipsoid.

### 2.3. Quantification of Extreme Heat Waves and Spatial Mapping

In this study, we used the hourly air temperature data of 20 national weather stations and 205 automatic weather stations for the past 10 years (2011–2020). This is the expanded dataset with the largest number of stations and the most abundant data. Based on hourly temperature data, the daytime (6:00–19:00) and nighttime (19:00–6:00) data in the past 10 years were sorted from small to large, and the 90% quantile was taken as the high temperature threshold. The cumulative hours of extreme heat waves at each station exceeding 33.1 °C during the daytime, during the summer (June to August), from 2011 to 2015 (12th Five-Year Plan) or from 2016 to 2020 (13th Five-Year Plan), were calculated, referred to as $EHW_{125d}$ and $EHW_{135d}$. The cumulative number of hours over 27.9 °C at night at each station was calculated, referred to as $EHW_{125n}$ and $EHW_{135n}$. Land cover (cropland, woodland, grassland, shrubland, wetland, water, urban land, and bare land), population density, DEM, and GDP, as the predictors, which affected the spatial distribution of high temperature, were selected [24,25]. Then, the extreme heat waves interpolation models were established based on urban surface land features as follows:

$$EHW_n = f(DEM, land_n, pop, GDP) \qquad (1)$$

where DEM is the elevation of the weather station, $Land_n$ represents the area of each land use type within the optimal buffer width of the land use type, pop is the population density of the site's location, GDP represents the gross domestic product, $EHW_n$ represents $EHW_{125d}$, $EHW_{125n}$, $EHW_{135d}$, or $EHW_{135n}$.

To obtain the spatial scale with the strongest correlations between various factors and $EHW_n$, 225 MSs were set with 100 buffer widths ranging from 1 to 100 km, and the land cover grid data were cropped with the vector files of the buffer. Using the spatial analysis function of ArcGIS, areas of 8 land cover types in 100 buffer zones of approximately 225 MSs were obtained (8 × 100 = 800 variables). DEM, GDP and population density data were point data (3 variables). In summary, 803 predictor variables were prepared.

The Pearson correlation coefficient is used to calculate the type of linear relationship between two variables (positive, negative, none) and the strength of this relationship (weak, moderate, strong). We used Pearson correlation coefficient to analyze the correlation between each variable and EHWn. Since the correlation coefficients between shrubland, wetland and water areas within the 1–100 km buffer zone and $EHW_n$ were not high, these variables were removed. Then, the buffer width with the highest correlation coefficient with $EHW_n$ was determined among the 100 buffer widths for the 4 land types of cropland, woodland, grassland and urban land. Stepwise linear regression was performed with $EHW_n$ as dependent variables, and cropland, woodland, grassland, urban land, population density,

DEM, and GDP as independent variables. Then, the multiple linear regression equations are obtained, which were the extreme heat waves interpolation models of $EHW_{125d}$, $EHW_{125n}$, $EHW_{135d}$, and $EHW_{135n}$ (see Supplementary Materials).

Based on the above multiple regression models, we interpolated the $EHW_n$ data of the kilometer grid cells using daytime and nighttime data from 225 stations, combined with resolution land use data of 30 m. First, 1 km $\times$ 1 km regular grid cells and grid points were generated for the study area; we calculated the value of the independent variable and the value of the result variable for each grid point and then we assigned the values of grid points to the grid to obtain the spatial distribution maps of daily/night extreme heat waves during the 12th and 13th Five-Year plans with a resolution of 1 km.

## 3. Result Analysis

### 3.1. Performance Evaluation of the Interpolation Models for Cumulative Hours of Urban Extreme Heat Waves

We used $EHW_{125d}$, $EHW_{125n}$, $EHW_{135d}$, and $EHW_{135n}$ from 225 MSs to correlate with the area of each land-use type within the 1–100 km buffer zone (Figure 2). With the change of buffer width, the cropland area in the buffer zone was positively correlated with $EHW_{125d}$, $EHW_{125n}$, $EHW_{135d}$, and $EHW_{135n}$. When the buffer width was 0–50 km, the absolute value of the correlation coefficient was less than 0.4, which was a weak correlation. When the buffer width was 50–100 km, the absolute value of the correlation coefficient was greater than 0.4 and less than 0.6, which was a moderate correlation. The area of woodland in the buffer zone was negatively correlated with $EHW_{125d}$, $EHW_{125n}$, $EHW_{135d}$, and $EHW_{135n}$. When the buffer width was 0–60 km, the absolute value of the correlation coefficient was greater than 0.6 and less than 0.8, which was a strong correlation. When the buffer width was 60–100 km, the absolute value of the correlation coefficient is greater than 0.4 and less than 0.6, which was a moderate correlation. The area of grassland in the buffer zone was negatively correlated with $EHW_{125d}$, $EHW_{125n}$, $EHW_{135d}$, and $EHW_{135n}$. When the buffer width was 0–40 km, the absolute value of the correlation coefficient was greater than 0.4 and less than 0.6, which was a moderate correlation. When the buffer width was 40–100 km, the absolute value of the correlation coefficient was greater than 0.6 and less than 0.8, which was a strong correlation. The area of urban land in the buffer zone was positively correlated with $EHW_{125d}$, $EHW_{125n}$, $EHW_{135d}$, and $EHW_{135n}$, and the absolute value of the correlation coefficient was greater than 0.6, which was a strong correlation. The area of shrubs, wetlands, and water in the buffer zone and $EHW_{125d}$, $EHW_{125n}$, $EHW_{135d}$, and $EHW_{135n}$ were sometimes positively correlated and sometimes negatively correlated, which may be related to the area and distribution pattern of shrubs, wetlands, and waters in Beijing.

By stepwise multiple regression, we obtained the multiple linear regression equations for $EHW_{125d}$, $EHW_{125n}$, $EHW_{135d}$, and $EHW_{135n}$ (see Supplementary Materials), which showed that urban land and DEM were predictors that enter into each resulting equation. This indicated that urban land and DEM were the variables most associated with $EHW_{125d}$, $EHW_{125n}$, $EHW_{135d}$, and $EHW_{135n}$ compared to other variables. The significance values of the four multiple linear regression equations by F-test ANOVA are all less than 0.01, indicating that the equations are highly significant. $R^2$ are 0.72 ($EHW_{125d}$), 0.77 ($EHW_{125n}$), 0.73 ($EHW_{135d}$), and 0.74 ($EHW_{135n}$), respectively, indicating that the fitting degree of the model is good. The fitting quality for nighttime ($EHW_{125n}$ and $EHW_{135n}$) is better than that for daytime ($EHW_{125d}$ and $EHW_{135d}$).

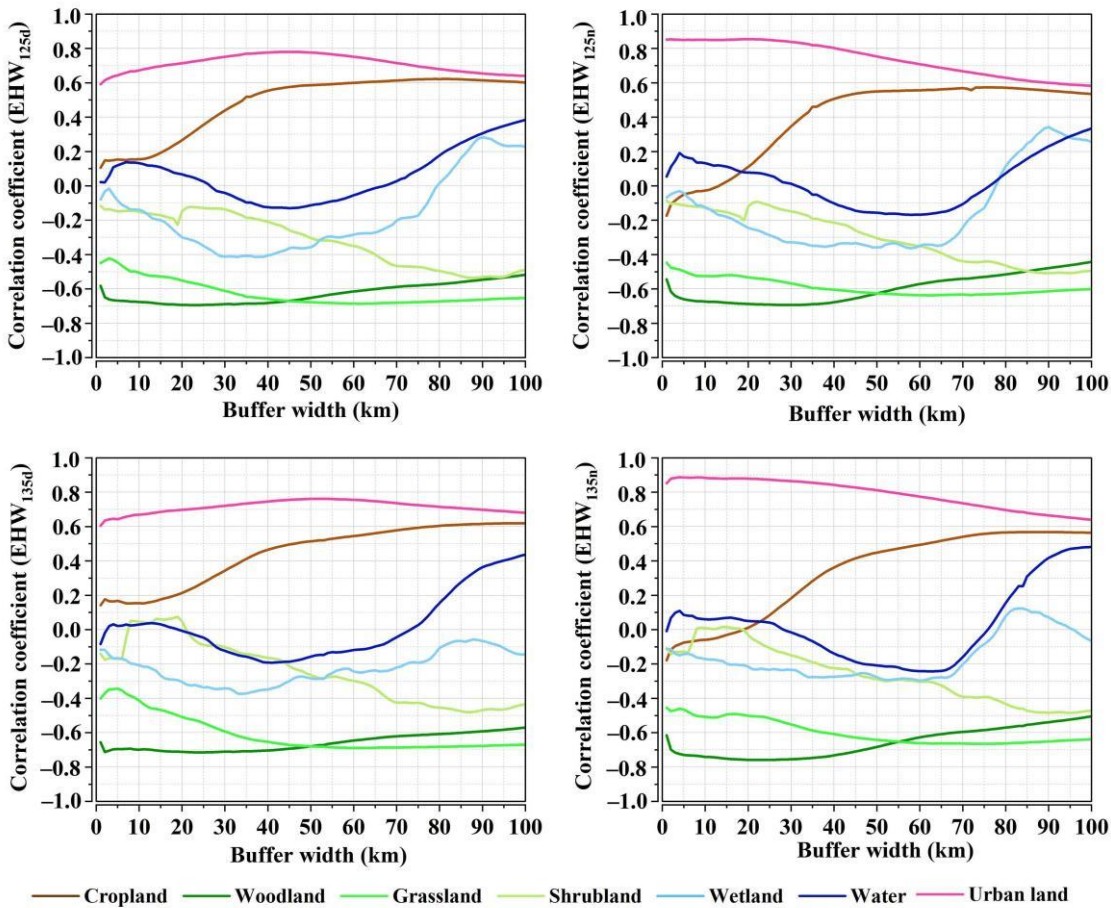

**Figure 2.** Correlation coefficients between the area of each land-use type in the 1–100 km buffer zones of the 225 MSs with $EHW_{125d}$, $EHW_{125n}$, $EHW_{135d}$, and $EHW_{135n}$. ($EHW_{125d}$ represents the cumulative hours of daytime extreme heat waves during the 12th Five-Year Plan period, $EHW_{125n}$ represents the cumulative hours of nighttime extreme heat waves during the 12th Five-Year Plan period, $EHW_{135d}$ represents the cumulative hours of daytime extreme heat waves during the 13th Five-Year Plan period, and $EHW_{135n}$ represents the cumulative hours of nighttime extreme heat waves during the 13th Five-Year Plan period).

### 3.2. Spatial Pattern Analysis of Cumulative Hours of Extreme Heat Waves during the Day/Night

The spatial distributions of $EHW_{125d}$, $EHW_{125n}$, $EHW_{135d}$, and $EHW_{135n}$ all show the pattern "high inside and low outside; high in the south and low in the north" (Figure 3), with a circle-layer structure that diverges from the central city to the suburbs. We resampled the raster map with a resolution of 1 km to the township scale and calculated the cumulative hours of extreme heat waves for each township. Then, divided the accumulated hours of extreme heat waves into 10 levels (Figure 3), where levels 1, 2, and 3 are low levels; levels 4, 5, and 6 are medium levels; levels 7, 8, and 9 are high levels; and level 10 is very high. Counted the number of townships that belong to the cumulative hours of extreme heat waves at different levels (Table 1). In 2011–2015, during the day, 68 townships were at low levels, 236 towns were at medium levels, and 27 towns were at high levels. At night, 94 townships were at low levels, 107 townships were at medium levels, and 130 townships were at high levels. In 2016–2020, during the day, 33 townships were at low levels, 35 towns were at medium levels, 102 towns were at high levels, and 161 towns were at the very high level. At night, 33 townships were at low levels, 38 townships were at medium levels, 129 townships were at high levels, and 131 townships were at the very high level. Compared with 2011–2015, from 2016 to 2020, there were 95 more high-level townships and 161 more extremely high-level townships in daytime, one less high-level townships and 131 more extremely high-level townships at night. This is because temperatures generally

increased between 2011 and 2020, and daytime temperatures increased even more. This is due to the general increase in temperature between 2011 and 2020, so the number of townships that are at extremely high-level during the daytime and nighttime has increased. In addition, the temperature rises more during the daytime compared to the nighttime, so the number of townships at extremely high-level during the daytime increases more.

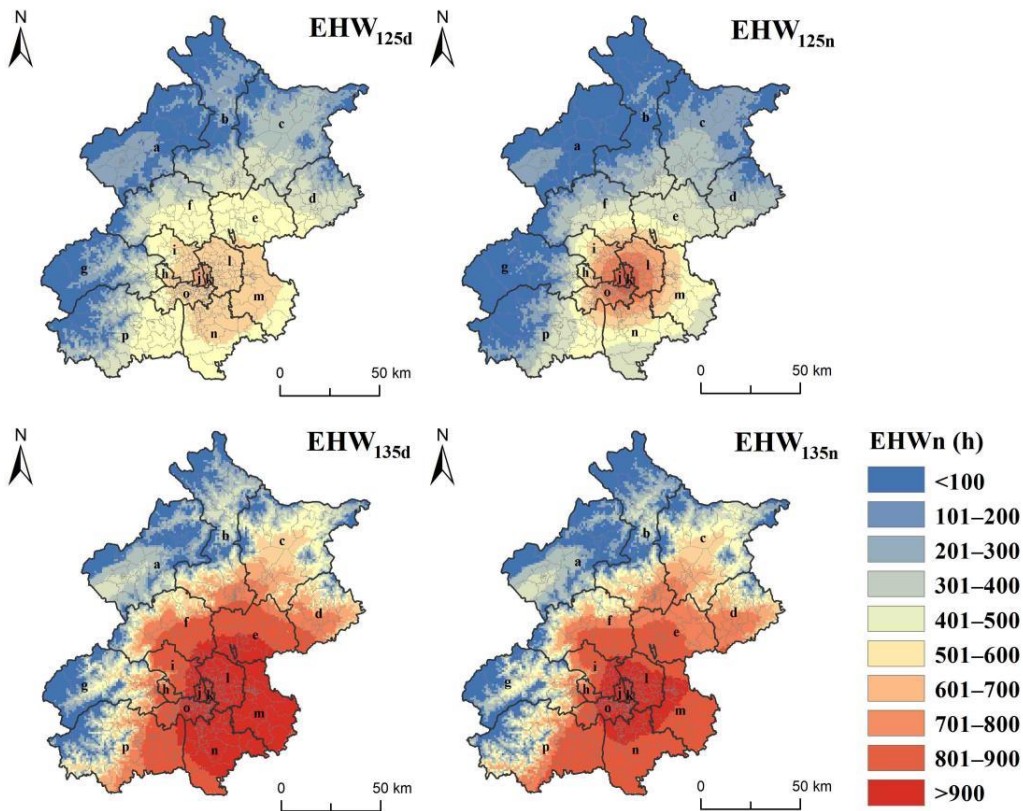

**Figure 3.** Spatial distribution maps of $EHW_{125d}$, $EHW_{125n}$, $EHW_{135d}$, and $EHW_{135n}$ in Beijing with a resolution of 1 km. ($EHWn$ represents $EHW_{125d}$, $EHW_{125n}$, $EHW_{135d}$, or $EHW_{135n}$).

**Table 1.** The number of townships that belong to the cumulative hours of extreme heat waves at different levels.

|  | $EHW_{125d}$ | $EHW_{125n}$ | $EHW_{135d}$ | $EHW_{135n}$ |
|---|---|---|---|---|
| Level 1, 2, 3 | 68 | 94 | 33 | 33 |
| Level 4, 5, 6 | 236 | 107 | 35 | 38 |
| Level 7, 8, 9 | 27 | 130 | 102 | 129 |
| Level 10 | - | - | 161 | 131 |

*3.3. Comparison of Accumulated Hours of Day/Night Extreme Heat Waves for Different Land-Use Types*

The average values of $EHW_{125d}$, $EHW_{125n}$, $EHW_{135d}$, and $EHW_{135n}$ on the surface of each land-use type in the whole Beijing, Beijing's urban areas, and Beijing's suburbs, were obtained through the spatial overlay of the land-use type vector map with $EHW_{125d}$, $EHW_{125n}$, $EHW_{135d}$, and $EHW_{135n}$. It can be seen in Figure 4 that the order of the average of cumulative hours of extreme heat waves on the surface of each land-use type in Beijing during the 12th and 13th Five-Year Plans was: urban land > cropland > water > grassland > woodland. During the 12th and 13th Five-Year Plan period, the order of the average of cumulative hours of extreme heat waves on the surface of each land-use type in Beijing's suburbs was as follows: urban land > cropland > water area > grassland > woodland. During the 12th Five-Year

Plan period, the order of the average of accumulated hours of extreme heat waves on the surface of each land-use type in the urban area of Beijing during the day was as follows: urban land > water > woodland > cropland > grassland. Additionally, the order of the average of accumulated hours of extreme heat waves on the surface of various land-use types in urban areas of Beijing at night was as follows: urban land > water > cropland > woodland > grassland. During the 13th Five-Year Plan period, the order of the average of accumulated hours of extreme heat waves on the surface of each land-use type in the urban area of Beijing during the day was as follows: urban land > water > cropland > grassland > woodland; the order of the average of cumulative hours of extreme heat waves corresponding to the surface of each land-use type in Beijing urban area at night was: urban land > water > cropland > woodland > grassland.

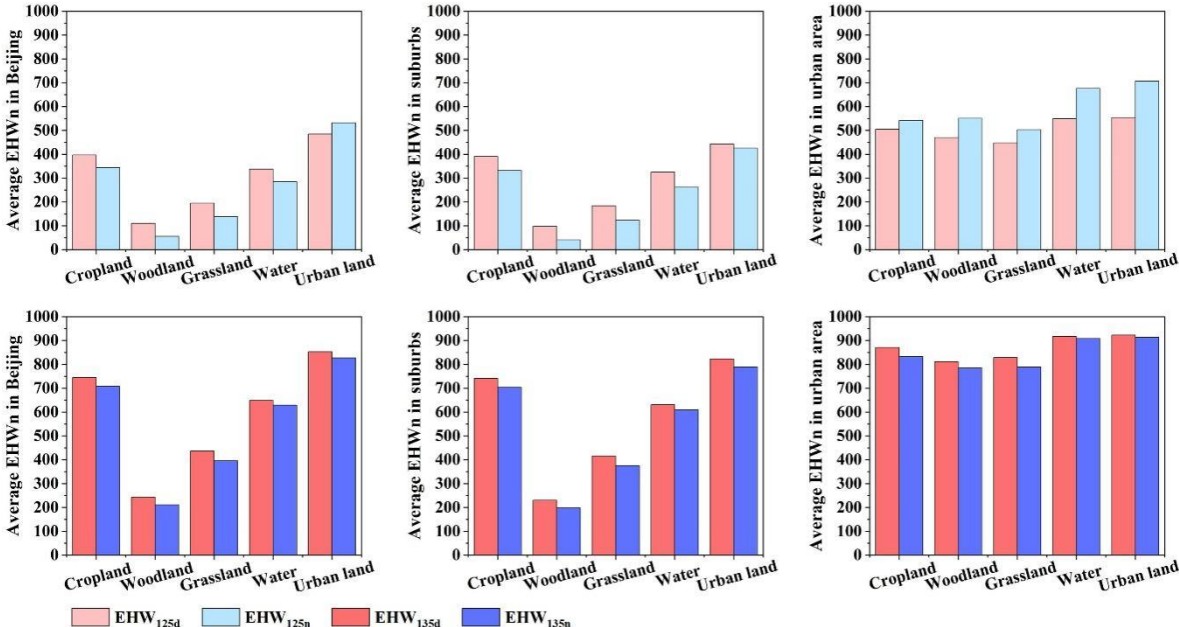

**Figure 4.** The average values of $EHW_{125d}$, $EHW_{125n}$, $EHW_{135d}$, and $EHW_{135n}$ for each land-use type in the whole Beijing, Beijing's urban areas, and Beijing's suburbs.

*3.4. Influence of Different Spatial Combination Patterns of Land-Use Type on the Cumulative Hours of Day/Night Extreme Heat Waves*

In order to further explore the impact of the spatial pattern of land-use types on the cumulative hours of day/night extreme heat waves, the land use pattern was represented by the proportions of the area occupied by the land-use types within a 1 km grid, and we analyzed their relationships with $EHW_{125d}$, $EHW_{125n}$, $EHW_{135d}$, and $EHW_{135n}$. First, the area proportion of each land-use type in each grid and the values of $EHW_{125d}$, $EHW_{125n}$, $EHW_{135d}$, and $EHW_{135n}$ in each grid were calculated. Then, the mean values of $EHW_{125d}$, $EHW_{125n}$, $EHW_{135d}$, and $EHW_{135n}$ of each land-use type at various proportions in the grid were counted at intervals of 20% for comparative analysis. From the change curves of the average values of $EHW_{125d}$, $EHW_{125n}$, $EHW_{135d}$, and $EHW_{135n}$ under different proportions of each land-use type (Figure 5), it can be seen that during the 12th and 13th Five-Year Plans, with the increase in the proportion of cropland in the grid, the average values of $EHW_{125d}$, $EHW_{125n}$, $EHW_{135d}$, and $EHW_{135n}$ of the grid also increased. Additionally, the increase rate during the day was greater than that at night, and the increase rate during the 13th Five-Year Plan period was greater than that during the 12th Five-Year Plan period. During the 12th and 13th Five-Year Plan period, with the increase in the proportion of woodland in the grid, the average values of $EHW_{125d}$, $EHW_{125n}$, $EHW_{135d}$, and $EHW_{135n}$ of the grid decreased, and the rate of decrease at night was greater than that during the day. The rate of decrease in the five-year period was greater than that in the 12th five-year period. During the 12th Five-Year Plan and the 13th Five-Year Plan (when the area proportion of

water in the grid was greater than 40%), as the proportion of water in the grid increased, the average values of $EHW_{125d}$, $EHW_{125n}$, $EHW_{135d}$, and $EHW_{135n}$ of the grid increased. During the 12th Five-Year Plan period, the rate of decrease at night was greater than that during the day, and during the 13th Five-Year Plan period, the rate of decrease during the day was greater than that at night. During the 12th and 13th Five-Year Plans, as the proportion of urban land in the grid increased, the average $EHW_{125d}$, $EHW_{125n}$, $EHW_{135d}$, and $EHW_{135n}$ of the grid also increased. Additionally, the increase rate at night was greater than that during the day, and the increase rate during the 13th Five-Year Plan period was greater than that during the 12th Five-Year Plan period.

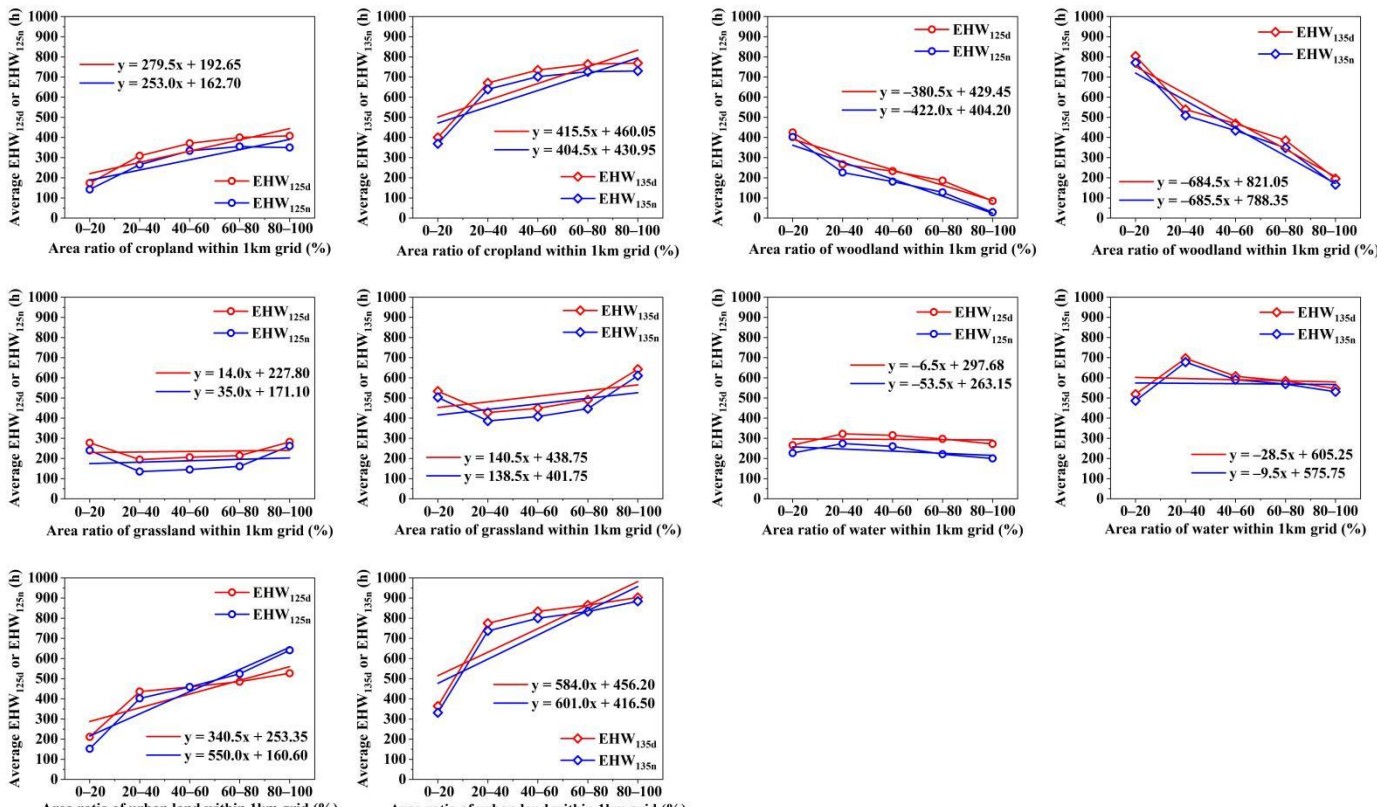

**Figure 5.** The EHWn changes ($EHW_{125d}$, $EHW_{125n}$, $EHW_{135d}$, and $EHW_{135n}$) with the area increasing of different land-use types. The area was represented as area ratio of each land-use type within the regular grids.

### 3.5. Mitigation Effects of Woodland and Water on Cumulative Hours of Day/Night Extreme Heat Waves in Beijing

The above analysis showed that the woodland and water area have key functions in alleviating the accumulated hours of extreme heat waves in Beijing, so the relationships between the areas of woodland and water and the accumulated hours of urban extreme heat waves were further analyzed. The grid-by-grid statistical analysis results of $EHW_{125d}$, $EHW_{125n}$, $EHW_{135d}$, and $EHW_{135n}$ (Table 2) show that the grids without woodland and water had the highest average values of EHWn, which were 444 ($EHW_{125d}$), 418 ($EHW_{125n}$), 849 ($EHW_{135d}$), and 812 ($EHW_{135n}$), respectively. The second highest average values of EHWn were for grids that were entirely water areas: 277 ($EHW_{125d}$), 202 ($EHW_{125n}$), 548 ($EHW_{135d}$), and 533 ($EHW_{135n}$). The average values of EHWn of the grids that were completely woodland were the lowest, which were 33 ($EHW_{125d}$), 0 ($EHW_{125n}$), 97 ($EHW_{135d}$), and 68 ($EHW_{135n}$). The average values of EHWn of the grids with woodland proportions greater than 50% were 114 ($EHW_{125d}$), 57 ($EHW_{125n}$), 251 ($EHW_{135d}$), and 218 ($EHW_{135n}$), and the average values of EHWn of grids with water proportions greater than 50% were 291 ($EHW_{125d}$), 226 ($EHW_{125n}$), 565 ($EHW_{135d}$), and 549 ($EHW_{135n}$). It can be found that,

regardless of the spatial distribution of woodland and water, the average values of $EHW_{125n}$ are smaller than that of $EHW_{125d}$, and the average values of $EHW_{135n}$ is smaller than that of $EHW_{135d}$. It can be seen that the woodland and water areas have obvious effects on alleviating the accumulated hours of extreme heat waves, and the mitigation effect at night is stronger than that in the daytime, and the mitigation effect of woodland is stronger than that of water area.

**Table 2.** Average values of $EHW_{125d}$, $EHW_{125n}$, $EHW_{135d}$, and $EHW_{135n}$ under different proportions of woodland and water area.

|  | $EHW_{125d}$ (h) | $EHW_{125n}$ (h) | $EHW_{135d}$ (h) | $EHW_{135n}$ (h) |
|---|---|---|---|---|
| Woodland (0%) & Water (0%) | 444 | 418 | 849 | 812 |
| Woodland (100%) | 33 | 0 | 97 | 68 |
| Water (100%) | 277 | 202 | 548 | 533 |
| Woodland (>50%) | 114 | 57 | 251 | 218 |
| Water (>50%) | 291 | 226 | 565 | 549 |

## 4. Discussion

This study analyzed daytime and nighttime cumulative hours of extreme heat waves under different land-use types in the representative metropolis of Beijing, and gave the high-resolution spatial mapping of day/night extreme heat waves cumulative hours. This study further explored influence of different land-use patterns on accumulated hours of day/night extreme heat waves. The results provided important reference for alleviating extreme heat waves in cities and for rational land planning. When cities expand rapidly, in addition to considering the expansion of building sites (due to their potential for extreme heat waves), other land use types (such as woodlands, grasslands, and water) should be considered to mitigate extreme heat waves.

Based on land use data with a resolution of 30 m and socioeconomic data with a resolution of 1km, this study established the urban day/night extreme heat waves interpolation model. Both the significance test and $R^2$ analysis showed that the model has good performance. The interpolation quality for nighttime was better than that for daytime. It can be a reference method for interpolating temperature data. In the final model, only the two to three most influential and contributing predictors were selected from a broad set of candidate predictors, mainly including DEM, population density, area of grassland and area of urban land within a specific buffer width. This suggests that DEM, population density, area of grassland, and urban land have the greatest correlation on urban extreme heat waves. It is worth noting that the land use patterns in different regions are different, so the obtained multiple regression interpolation models will also be different.

The spatial distribution of the cumulative hours of day/night extreme heat waves in Beijing is "high inside and low outside; high in the south and low in the north". In other words, the cumulative hours of extreme heat waves in the eight districts of Dongcheng, Xicheng, Chaoyang, Haidian, Fengtai, Shijingshan, Daxing, and Tongzhou were relatively high, while the cumulative hours of extreme heat waves in the remaining eight districts were relatively low. This is related to factors, such as land use, population density, DEM, etc. The area ratio of urban land in the eight districts of Dongcheng, Xicheng, Chaoyang, Haidian, Fengtai, Shijingshan, Daxing, and Tongzhou is 54.22%, the area ratio of woodland is 5.15%, the average population density is 4301 persons/km$^2$, and the average altitude is 44.83m. In the other eight districts, the proportion of woodland is 55.20%, the proportion of urban land is 12.84%, the average population density is 518 persons/km$^2$, and the average altitude is 450.43 m. These differences are important reasons for this spatial distribution pattern.

We simulated the spatial distribution pattern of accumulated hours of day/night extreme heat waves in Beijing in 2011–2015 and 2016–2020. In 2016–2020, the average annual cumulative hours of extreme heat waves were 95.06% (daytime) and 115.70%

(nighttime) higher than those in 2011–2015. This is related to natural climate change and human activities [26–28]. The WMO survey results show that 2011–2020 was the warmest decade on record. The warmest six years were 2015 and subsequent years, with 2016, 2019, and 2020 making up the top three. This is an important reason why the accumulated hours of extreme heat waves in 2015–2020 were much higher. Compared with 2011–2015, the average annual resident population in 2016–2020 increased by 3.54%, and the GDP increased by 64.41%. Compared with 2010, the urban land area by 2020 had increased by 55.43%. Human activities, such as population movement, urban land expansion, and fossil fuel burning, are also another important reason why the cumulative hours of extreme heat waves in 2016–2020 were much more than those in 2011–2015.

Our research shows that the order of land-use types based on the cumulative hours of extreme heat waves from different land types in Beijing were as follows: urban land > cropland > water > grassland > woodland. Urban land had the most cumulative hours of extreme heatwaves during the day and night, the woodland had the fewest cumulative hours of extreme heat waves during the day and night. Cropland and urban land can increase the cumulative hours of extreme heat waves, whereas woodland and water areas can reduce the cumulative hours of extreme heat waves. The effect of grassland on the cumulative hours of extreme heat waves was not significant. The role of cropland in enhancing the cumulative hours of extreme heat waves is greater during the day than at night; the role of urban land in enhancing the cumulative hours of extreme heat waves is greater at night than in the daytime, and the role of woodland in alleviating the cumulative hours of extreme heat waves is greater at night than in the daytime. This day–night difference is related to the complex surrounding environment and the differences in day–night specific heat capacities of the various land-use types [29–31]. Water has the highest specific heat capacity, followed by woodlands and grasslands with higher water content (woodlands have higher water content than grasslands) and least impermeable surfaces. Urbanization leads to reduced evaporation and wind. In addition, the lower albedo of impervious surface compared to vegetation increases daytime heat storage and enhances nighttime long-wave heat release. It should be noted that our research shows that grassland has the effect of enhancing the cumulative hours of extreme heat waves. As can be seen in the Figure 5, this result was mainly caused by the high cumulative hours of extreme heat waves in grids with grassland accounting for more than 80% of the area. Both 2010 and 2020, among the 15,818 grids in Beijing, there were only approximately 80 grids that had more than 80% grassland. Nearly 2/3 of the 80 grids were close to cultivated land or urban land. Environmental impacts lead to higher cumulative hours of extreme heat waves. It can be seen in the Figure 5 that the cumulative hours of extreme heat waves for grids with water areas less than 20% were significantly lower than those grids with larger water areas. This is because the water area of Beijing is too small, only 0.92% (2010) or 1.45% (2020). As a result, in 2010 and 2020, the proportions of grids with water area accounting for 0–20% were extremely high, 98.8% and 98.1%, respectively. The grids with water areas of 0–20% are numerous and continuously distributed, making them less susceptible to the surrounding environment. On the contrary, grids with water areas greater than 40% are small in number and scattered, and are easily affected by the surrounding environment, which makes the cumulative hours of extreme heat waves relatively high. However, when the proportion of the water area is greater than 40%, the cumulative hours of extreme heat waves is decreased significantly as the proportion of the water area continues to increase, which reflects the role of water area in relieving the cumulative hours of extreme heat waves. Different from water, the area of woodland in Beijing accounted for 44.74% (2010) and 45.08% (2020), and the distribution was concentrated. This difference between woodland and water in area and spatial pattern may lead to Beijing's woodland having a stronger mitigation effect on the cumulative hours of extreme heat waves relative to water.

## 5. Conclusions

The main conclusions of our study are summarized as follows:

1.  The urban day/night extreme heat waves cumulative hourly interpolation models were established; the correlations were highly significant at $p < 0.01$, both the significance test and $R^2$ analysis showed that the models have good performance, and the accuracy of interpolation was high;

2.  The spatial distribution of the cumulative hours of day/night extreme heat waves in Beijing is "high inside and low outside; high in the south and low in the north"; a circle structure radiates from the central city to the suburbs. During 2016–2020, the annual cumulative hours of extreme heat waves in the daytime and in the nighttime were 95.06% and 115.70% higher than those of 2011–2015;

3.  The order of land-use types for cumulative hours of land-surface extreme heat waves in Beijing was as follows: urban land > cropland > water > grassland > woodland. In most cases, the cumulative hours of extreme heatwaves during the day were greater than those at night for cropland, woodland, grassland, water, and urban land;

4.  The increase associated with urban land in the cumulative hours of extreme heat waves at night is greater than that in the daytime. The mitigation effects of woodland and water on the cumulative hours of extreme heat waves are stronger at night than in the daytime, and the mitigation effect of woodland is stronger than that of water.

**Supplementary Materials:** The following supporting information can be downloaded at: https://www.mdpi.com/article/10.3390/land11101786/s1.

**Author Contributions:** Conceptualization, F.W.; data curation, X.S. and H.Z.; formal analysis, X.S.; funding acquisition, F.W.; methodology, X.S. and F.W.; software, X.S.; supervision, F.W. and D.Z.; writing—original draft, X.S.; writing—review and editing, F.W. All authors have read and agreed to the published version of the manuscript.

**Funding:** This work is jointly supported by National Key Research and Development Program of China (2019YFC0507805) and the Strategic Leading Science and Technology Program of the Chinese Academy of Sciences (XDA20020202).

**Institutional Review Board Statement:** Not applicable.

**Informed Consent Statement:** Not applicable.

**Data Availability Statement:** The processed data, which were used to generate the figures and tables, are available upon request to the corresponding author.

**Acknowledgments:** We express our sincere appreciation to the anonymous reviewer for constructive comments.

**Conflicts of Interest:** The authors declare no conflict of interest. The funders had no role in the design of the study; in the collection, analyses, or interpretation of data; in the writing of the manuscript, or in the decision to publish the results.

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
