# Peer review of "Assessing the Spatial Variability of Daytime/Nighttime Extreme Heat Waves in Beijing under Different Land-Use during 2011–2020"

_land, doi:10.3390/land11101786_

Round 1
Reviewer 1 Report
Please refer to the attachment for my comments.
Best wishes

Reviewer 2 Report
In the article, heat waves in Beijing are analyzed for the time range of 2011–2020 with the use of data from 225 meteorological stations. Influence of several factors including land cover, population density, gross domestic product, elevation is estimated. It is shown that a large increase in both daytime and nighttime heat waves was observed during 2016–2020 in comparison with 2011–2015. This dangerous phenomenon is clearly demonstrated in Fig. 3. It was concluded that woodland and water have the maximal potential to weaken extreme heat waves.
The study was carried out at a high scientific level and its results are clearly presented. Comments are collected in the attached file, but they are mostly technical. More detailed explanations are desirable in several cases (see the remarks to L. 83, 94, 141). In my opinion, only minor corrections should be done before publishing the manuscript in Land.

Reviewer 3 Report
Dear Authors,
a) I don't fell that I am qualified enough to judge about English writing, but I must say that understanding what is written in this paper was a bit hard for me. So, I would suggest major revision as far as English language is concerned.
line 11:
current sentence: "Urban land-use changes impact surface air temperature, but the extent to which land use 11 affects extreme heat waves and their duration remain uncertain, and the mechanism of their day and night differences needs to be further explored"
proposed sentence: "... but the extent and their duration to which extreme heat waves affects land, remain uncertain"
line 13:
current sense: "Here, we study daytime/nighttime…"
proposed sentence: "This paper presents study about daytime/..."
line 21:
unclear sentence: "We found the strengthening effect of urban land on the cumulative time of extreme heat waves at night was greater than that in daytime, and woodland and water mitigated the effects of long extreme heat waves better during nighttime, with woodland showing stronger effects than water."
line 34:
current sentence: "Urban land use 34 expansion is closely related to urban high temperatures [5,6,7],..."
proposed sentence: "Usage of the urban land expansion is closely.."
line 45:
current sentence: "..and surface temperature [14,15], or using remote sensing..."
proposed sentence: "...usage of remote sensing..." or "or remote sensing use..."
b) instead of using "For example, references" in line 41, please use more proper syntax.
For example: Studies are made about the impact of urban..
line 56:
instead of: "We used 225 meteorological stations data", use more clear way to describe what area is object of measurements, by what and from who:
"Data measured by 225 meteorological stations in Bejing area used in this study and are provided by the Bejing Meteoro... Disaster Center."
line 99: "The population unit is person/km2" - population density is proper term
c) when stating that something is used, stress why is such selection best for this case and how something is actually done. Also be precise when stating your opinion.
line 86:
"Strict quality control measures..." - explain process, at least in few sentences. From current context, reader can just conclude that this is not made by "guessing"
Also in same paragraph, different kind of measurement stations are used. Please provide some basic technical information about each station type.
For example, AWS stations are made by "some manufacturer, model". Sensors used for measuring "what (air temperature, relative air humidity?...)" are developed "by who". Without this information reader can not determine what grade of sensors are used and what is expected quality of data provided by such sensors are used in this study.
line 142:
"The Pearson correlation coefficient was used to analyze" - stress why is this right selection for this study (I don't propose that it is not, just stress for reader why, since we are not experts in every field, but many our fields interpolates)
line 313:
"The results provided important reference for alleviating extreme heat waves in cities and for rational land planning" - stress some of actual examples of how rational land planning should be done
line 374:
"The urban day/night extreme heat waves cumulative hourly interpolation models 374 were established with good performance" - what does good performance means? Short calculation time,...?
d) when equations are used, describe each variable name, what does it actually represents in paragraph describing it:
EHWn = f (dem, landn, pop,GDP)
dem - is what
landn - is what
pop - is what
GDP - is what
This way, reader can conclude what each variable represents, but it should be always clearly written instead.
I firmly believe that this paper has good potential, but in my opinion it should be heavily modified. I hope that this suggestions will not stop you in finishing what you have already started.
Best regards.
Round 2
Reviewer 1 Report
It is so nice to see the authors sincere effort to address all the comments.
Thank you.
However, I still believe that four of my comments were not addressed well. Hence, I will be glad, if authors reconfirm those comments.
Looking forward.
Best wishes

Reviewer 3 Report
Dear Authors,
Thank you for accepting suggestions and making this paper better.
Keep up good work.
As far as I am concerned, this paper can be published.
Regards
Author Response
Thank you for your suggestions to make our manuscript better.